# Subduction initiation triggered the Caribbean large igneous province

Nicolas Riel [1,2] ✉, João C. Duarte [3], Jaime Almeida [3], Boris J. P. Kaus [1,2], Filipe Rosas [3], Yamirka Rojas-Agramonte [1,4] & Anton Popov[1]

Subduction provides the primary driving force for plate tectonics. However, the mechanisms leading to the formation of new subduction zones remain debated. An example is the Lesser Antilles Arc in the Atlantic. Previous initiation mechanisms have implied the transmission of subduction from the Pacific Ocean or the impact of a plume head. Here, we use geodynamic models to simulate the evolution of the Caribbean region during the Cretaceous, where the eastern Pacific subduction triggered the formation of a new subduction zone in the Atlantic. The simulations show how the collision of the old Caribbean plateau with the Central America margin lead to the formation of a new Atlantic subduction zone by polarity reversal. The results further show how subduction renewal on the back of the old Caribbean plateau (present-day Central America) resulted in a major mantle flow reorganization that generated a subduction-induced plume consistent with the formation of the Caribbean Large Igneous Province.

Subduction is the process by which cold oceanic lithosphere is recycled back into the mantle. Together with plumes and oceanic floor generation at mid-ocean ridges they form the back-bone of plate tectonics, and their dynamics drive supercontinent amalgamation and break-up. While oceanic spreading and subduction dynamics are now fairly well understood, the transition from a passive spreading ocean to an active subduction zone together with its geodynamic consequences remain poorly investigated[1]. This is mainly because there are very limited examples of ongoing subduction initiation on Earth[2] and our rheological understanding of passive margins suggests that they remain difficult to break and initiate subduction[3–6] unless mechanical weakening of the margin is achieved[7] and/or external forces are applied[1,2]. Yet, along the Atlantic passive margins active subduction has initiated in the Caribbean and Scotia sea. In the Caribbean, subduction transfer from the Pacific to the Atlantic coincides with the formation of a large igneous province (LIP). While the late Cretaceous voluminous magmatic and volcanic activity recorded in the Caribbean (CLIP) is related to mantle plume activity with a debated link to the Galapagos hotspot[8,9] and with potential mantle temperature reaching up to 1630 °C[10–12], the relationships between the plume formation and the subduction dynamics of the Caribbean region remain poorly understood. Understanding the geodynamic conditions that led to the transfer of subduction from Pacific to Atlantic in the Caribbean region is therefore of primary importance, to unravel the formation of the Caribbean large igneous province and to better understand the processes of subduction transfer.

The Caribbean region constitutes a natural laboratory where long-term plate-tectonics over 140 Myr resulted in the transference of a subduction zone from the Pacific to the Atlantic margin, while producing the voluminous magmatic activity that accounts for the Cretaceous Caribbean Large Igneous Province (CLIP)[11,13,14], now located in the center of the Caribbean plate (Fig. 1). Although numerous, geochronological[15,16], geochemical[11,14,16–18], and plate reconstructions[9,19,20] studies helped to better constrain the Cretaceous geodynamic evolution of the Caribbean region, no geodynamic consensus has been reached yet. This is mainly because of the efficient recycling of paleo-oceanic plates by subduction, the lack of physical constraints of existing geodynamic reconstructions and the highly complex plate tectonic setting of the region. Indeed, the geological setting of Central America is characterized by the interaction of several

[1]Institute of Geosciences, Johannes Gutenberg-University, Mainz, Germany. [2]Terrestrial Magmatic Systems (TeMaS) Research Center, Johannes Gutenberg-University, Mainz, Germany. [3]Instituto Dom Luiz, Faculty of Sciences, University of Lisbon, Lisbon, Portugal. [4]Institute of Geosciences, University of Kiel, Kiel, Germany. ✉e-mail: nriel@uni-mainz.de

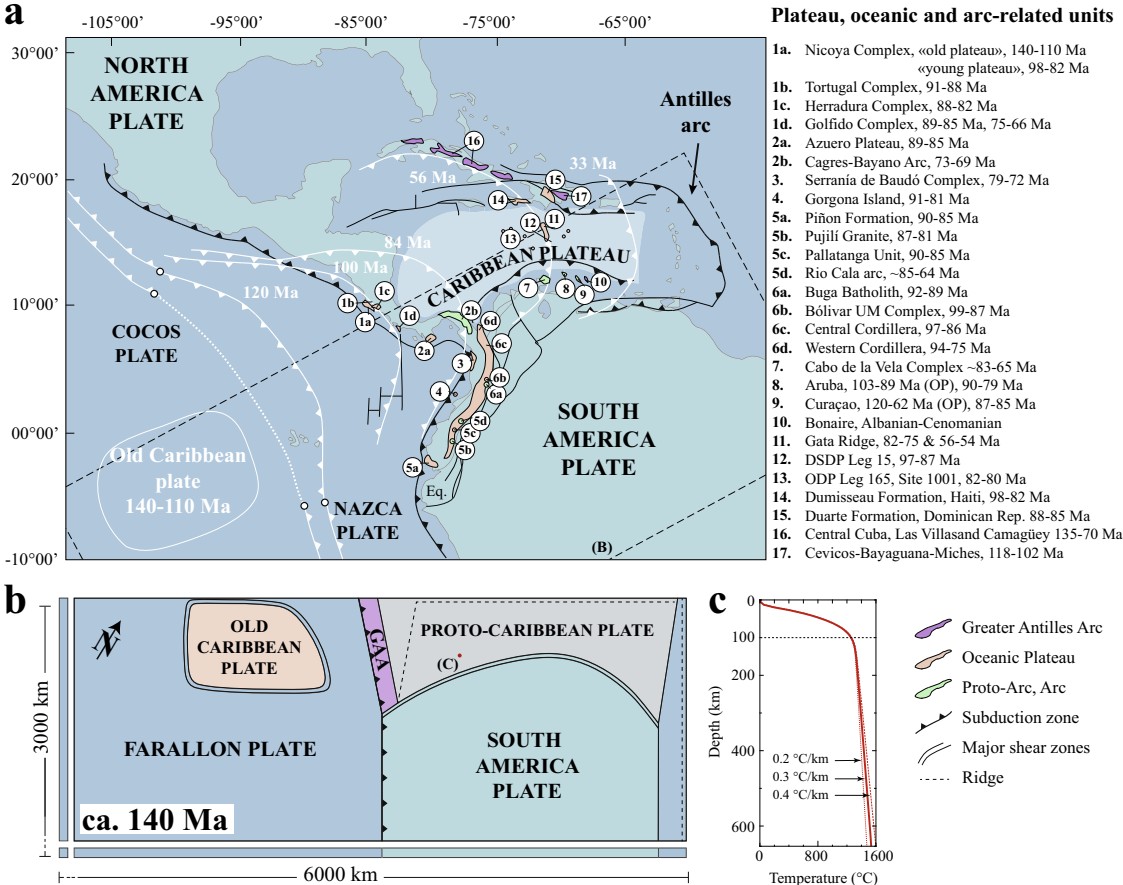

**Fig. 1 | Simplified geological map of the Caribbean. a** actual and reconstructed position of the Caribbean plateau after[31]. Plateau, oceanic and arc-related units modified after[14]. Age references: 1a[13,36,38], 1b[38,46], 1c[38], 1d[38,47,48], 2a[49–52], 2b[51–53], 3[54], 4[17,55–58], 5a[59–61], 5b[62–64], 5c[63,65], 5d[63,64,66–69], 6a[70], 6b[17,70,71], 6c[54,72], 6d[37,54,72–74], 7[75], 8[76–78], 9[18,37,57,78], 10[78], 11[37], 12[37,39], 13[79,80], 14[18,37], 15[81–83], 16[15], 17[26]. GAA, Greater Antilles Arc. **b** map view of the reference model setup. **c** initial temperature profile. Heavy line: reference initial temperature gradient of 0.3 ℃/km for the upper mantle. Dashed lines, other investigated initial upper mantle temperature gradients (0.2 and 0.4 ℃/km).

lithospheric plates around the Caribbean plateau. The Caribbean plateau itself is regarded as oceanic lithosphere made of a 15–20 km overthickened oceanic crust[21]. Presently, the Caribbean plateau is bounded by the eastward subduction of the Cocos plate to the west and the westward subduction of the Atlantic plate below the lesser Antilles arc to the east. To the north, the Caribbean plateau is separated from the North American plate by a lithospheric transform fault zone and to the South the Caribbean plateau underthrusts the continental South American plate. Studies suggest that the Caribbean plateau is composite and its build-up results from two long-lived events of magmatic/volcanic construction: an old one at 140–110 Ma with a distinctive plume-related signature[11,14,22] and the younger Caribbean Large Igneous Province (CLIP) event between 97 and 70 Ma with a gradual change from plume to plume-subduction hybrid magmatic signature over time[11,13,14]. The first early Cretaceous magmatic event at the origin of the old Caribbean plateau is thought to have been less voluminous with respect to the second late Cretaceous CLIP event[11,14]. In this study, we refer to the proto-Caribbean plate (Fig. 1b) as the former oceanic plate (Atlantic-derived) separating North and South America continental plates. Furthermore, we refer to the first magmatic stage of Caribbean crust thickening as the old Caribbean plateau, while the CLIP event emplaced through the old Caribbean plateau forms the bulk of the present day Caribbean plateau[11,14]. This distinction is important as our simulations are initiated with the old plateau already formed, while the tectono-magmatic event that originates the CLIP is investigated.

Yet, the geographic origin of the old plateau and the geodynamic conditions leading to the CLIP event are still debated. Two endmember models have been proposed. In the first scenario, the old Caribbean plateau formed close to its present-day position while the westward subduction of the proto-Caribbean plate, was established by 110 Ma on the eastern edge of the old plateau[11,14]. Several studies, however, suggest that the westward subduction of the proto-Caribbean plate initiated 35 Myr earlier, at ca. 135 Ma[15,19,20,23–27]. Subsequently, the plume head at the origin of the CLIP event was emplaced through the old plateau at ca. 100 Ma. Plume-related weakening of the lithosphere and edge downwelling has been proposed to have initiated north-east-directed subduction of the Farallon plate below the western to south-eastern edge of the Caribbean plateau[11,28,29]. The second model, supported by recent plate reconstructions[19,20], places the formation of the Caribbean old plateau in the Pacific ocean away from the American margin[20,22]. Subduction of the Farallon plate below the Americas led to the collision of the old Caribbean plateau with the proto-Caribbean plate, resulting in the westward subduction initiation of the proto-Caribbean plate below the eastern edge of the old plateau. This drove the eastward motion of the Caribbean plateau between North and South America and ultimately resulted in the migration of the subduction zone into the Atlantic (Figs. 1 and 2). Collision of the old Caribbean plateau also resulted in the formation of a slab window. While several studies support the idea that the CLIP could have originated as a consequence of a slab window[8,17], other studies have

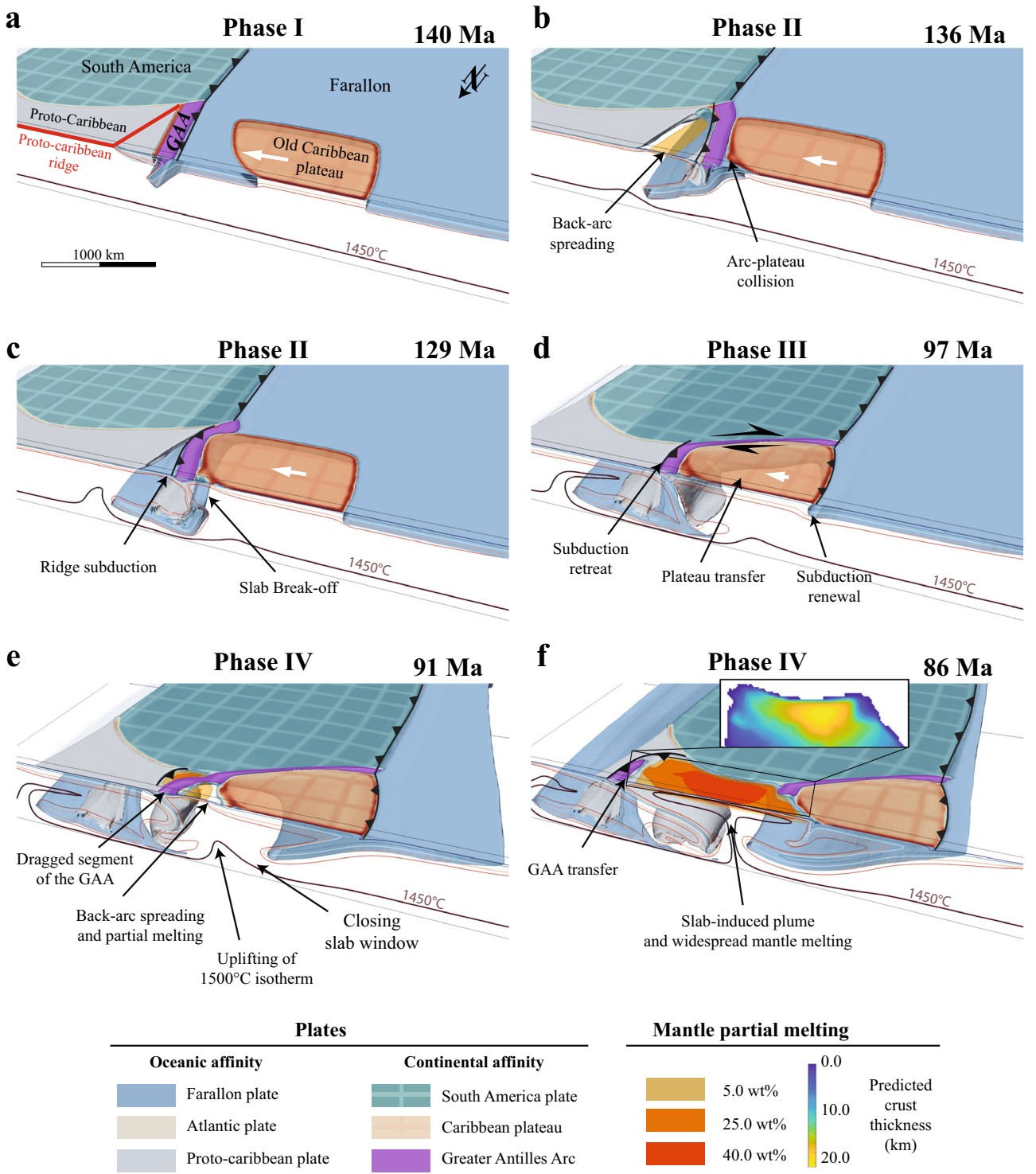

**Fig. 2 | Reference simulation results. a** initial conditions. **b** collision between old Caribbean plateau and Greater Antilles Arc (GAA). During this phase east-directed subduction of the Farallon locally ceases and west-directed subduction of the proto-Caribbean plate initiates (Polarity-reversal subduction initiation). **c** plateau transfer phase and mantle window opening. **d** subduction initiation of Farallon plate at the western edge of the Caribbean plateau. **e** Triggering of a subduction-induced plume and partial transfer of the Greater Antilles Arc onto the rolling back proto-Caribbean plate. **f** fully developed mantle plume and widespread partial melting of the sub-lithospheric mantle reaching up to 40 wt%. Melt content is post-processed using mantle melting parameterization[32]. The resulting thickness of the crust is estimated by vertically integrating the total volume of predicted partial melt.

shown that a slab window origin for the CLIP event is unlikely as the melt production would have been too low and the chemical signature of the CLIP reflects a dry and therefore deeper origin[24]. Whether or not the CLIP event resulted from plume activity[11,14], slab window[17] or a combination of both, the magmatic signature of the CLIP exhibits a clear plume-related anomalous hot mantle source[10,11] that gradually hybridizes with subduction-related magmatism[11,13,14].

Here, we use large-scale 3D geodynamic simulations with free surface to study the Pacific origin scenario of the old Caribbean plateau[9,19,20]. We use the finite difference code LaMEM[30] to solve the

energy, momentum, and mass conservation equations. We test several initial geometries and sizes for the old plateau and explore the role of the mantle rheology and initial mantle temperature profile - to investigate the geodynamic mechanisms responsible for the old Caribbean plateau collision and transfer, subduction propagation from Pacific to Atlantic and formation of the CLIP. Based on clear evidence of long-lasting plume activity recorded in the Caribbean region throughout most of the Cretaceous[11,13,14] (140–110 and 100–70 Ma) we choose to use the Boussinesq approximation (deactivate adiabatic heating) in our simulations, i.e., we assume that the mantle of the Caribbean region was anomalously hot and buoyant due to plume activity during the entire modelled time window (140–70 Ma). Furthermore, we show in Supplementary Information that using the extended Boussinesq approximation (activate adiabatic heating) yield very similar results.

We simplify the Caribbean plate configuration at ca. 140 Ma as symmetric with the axis of symmetry being the paleo-proto-Caribbean ridge (Fig. 1b). We choose to reduce our model domain to the South America side as plate reconstructions show that collision of the Caribbean plateau occurred along the north-western margin of the South American plate[9,19,20] as illustrated by the accretion of oceanic terranes (Fig. 1a). The tested initial configuration for the proto-Caribbean/Farallon boundary at ca. 140 Ma is inspired by recent geodynamic reconstructions[19,20,25]. We assume a north to south continuous subduction along the American margin with the Farallon plate initially subducting below the Greater Antilles Arc[19,20] (Fig. 1b). The initial presence of Greater Antilles Arc in our setup contradicts the evidence that the arc started to form 10 Ma later, as a response of the westward proto-Caribbean subduction[15,31]. However, we do prescribe its presence from the start to be able to track its subsequent tectonic evolution. Moreover, we notice that the onset of the old plateau magmatism at ca. 140 Ma[11,14] happened very shortly before the initiation of the proto-Caribbean plate subduction at ca. 135 Ma[15,19,20,23–27]. In this regard, it cannot be excluded that these two events are related and that the proto-Caribbean plate subduction may have been initiated by plume activity and edge downwelling[28,29].

We performed systematic simulations to test the robustness of our modelling results to variations in different parameters, including mantle temperature profiles and plateau geometries (see Supplementary Information). Details about the mathematical formulation used in LaMEM and the modelling strategy to study the Caribbean system are presented in appendices 4.1 and 4.2, respectively. It should be noted that no interior kinematic constraints are prescribed in our simulations, and thus the geometric and kinematic evolution of the system self-consistently results from the buoyancy and resisting viscous forces between the plates and the surrounding mantle.

## Results

All simulations yielded a similar first order geodynamic evolution that can be summarized in 4 distinct phases (see the reference model in Fig. 2 and Supplementary Movie 1). During phase I (Fig. 2a), the ongoing subduction of the Farallon plate below the Central and South American margin drives the motion of the old Caribbean plateau towards the Farallon trench (Fig. 2a). Phase II is marked by the collision between the Caribbean plateau and the proto-Caribbean margin (Fig. 2b). Plateau collision triggers slab break-off of the Farallon plate, westward polarity reversal subduction initiation of the proto-Caribbean plate below the Greater Antilles Arc (Fig. 2c), and the formation of a mantle window connecting the Farallon, and South America asthenospheres (Figs. 2c and 3). Phase III is defined by subduction rollback of the proto-Caribbean plate and eastward drag of the overriding old Caribbean plateau (Fig. 2d). During this stage, a strong dextral transpressive motion is recorded at the southern edge of the plateau between South America and the southern part of the Greater Antilles Arc being accreted against the northern South American margin (Fig. 2c). Phase IV is characterized by subduction initiation on

the western edge of the plateau and coeval back-arc opening between the Caribbean plateau and the proto-Caribbean trench (Figs. 2e, 3a, b). This results in the partial transfer of the Greater Antilles Arc from the eastern edge of the Caribbean plateau onto the rolling back proto-Caribbean plate (Figs. 2f, 3b). Meanwhile, subduction initiation of the Farallon subduction is accompanied by a major change of the proto-Caribbean underlying mantle flow (Fig. 3a, b). As the Farallon plate sinks below the Caribbean plateau, the closing mantle window forces the mantle to flow upward, effectively generating an upper-mantle scale upwell of hot and buoyant material (Fig. 3c) which we coin subduction-induced plume.

The systematic simulations presented in Supplementary Information show how the differences in the initial temperature/geometry/mantle rheology influence second order geodynamic features such as: partial melting and estimated excess volume of magma; the fragmentation of the old plateau and the geometry of the subduction-induced plume head (e.g. simulation CP.7, Supplementary Fig. 12); the geometry of the central America margin after the old plateau transfer (e.g. simulation CP.6, Supplementary Fig. 11); and the partial transfer of the Greater Antilles Arc onto the Caribbean plateau during the proto-Caribbean roll back (e.g., simulations LP.1, MP.1 and CP.3, Supplementary Figs. 4, 14 and 8, respectively).

## Discussion

The first modeled tectonic event is the collision between the plateau and the proto-Caribbean plate (Phase II, Fig. 2b) which occurs at ca. 135 Ma, interrupting subduction of the Farallon plate below the proto-Caribbean and subsequently followed by westward subduction initiation of the proto-Caribbean plate and coeval transfer of the plateau along the northwestern South American margin (Fig. 2c, d). This is in agreement with available studies indicating that westward subduction of the proto-Caribbean plate initiated at 135–125 Ma[15,19,20,23–27]. At 105 Ma, our simulations predict that the slowdown of the Caribbean plateau transfer triggers subduction initiation of the Farallon plate at the western edge of the old plateau (Fig. 4). Recent tectonic reconstructions[20] infer that subduction on the western edge of the plateau was initiated ca. 100 Ma, roughly 10–15 Ma before the beginning of the CLIP event. As subduction initiates on the western side of the Caribbean plateau (phase IV) the closure of the mantle window from 97 Ma onwards triggers a key reorganization of the mantle flow (Fig. 3). The sinking of the Farallon plate squeezes the mantle against the retreating proto-Caribbean slab (Figs. 3, 4), forcing the roll back of the proto-Caribbean plate, back-arc spreading, and inducing the formation of an upper-mantle plume ascending at the eastern edge of the closing slab window (Figs. 3c and 4). Using parameterized dry mantle melting[32], we find that partial melt content reaches a maximum of 40 wt% in the central region of the plume-head (Fig. 2f) consistent with geochemical estimates of 32 wt%[24,33]. Calculation of the generated crust thickness ranges from 10 to 22 km and is also consistent with available data on the present-day Caribbean plateau crust thickness[21]. Moreover, the computed excess magma volume (oceanic crust ≥10 km) yields $5.5 \times 10^6$ km$^3$, similar to estimated $4.4 \times 10^6$ km$^3$ for the CLIP event[34]. Note that the computed value of $5.5 \times 10^6$ km$^3$ is an upper limit as our calculation assume batch melting and not incremental melting that is known to decrease the total degree of melting by up to 25%[35], so the slightly higher melting value is consistent with the observations.

The modeled timing of the subduction-induced plume (Fig. 2) agrees relatively well with evidence showing that the CLIP event started at ca. 97 Ma[17,18,36–39]. However, in the reference simulation (Fig. 2), the initiation of the CLIP event occurs at ca. 90 Ma, nearly 7 Myr later than in the reconstructions. This difference is smaller for simulations in which the segment of the proto-Caribbean ridge parallel to the Farallon trench (Fig. 1b) is not included. This results in a 8 Myr decrease of

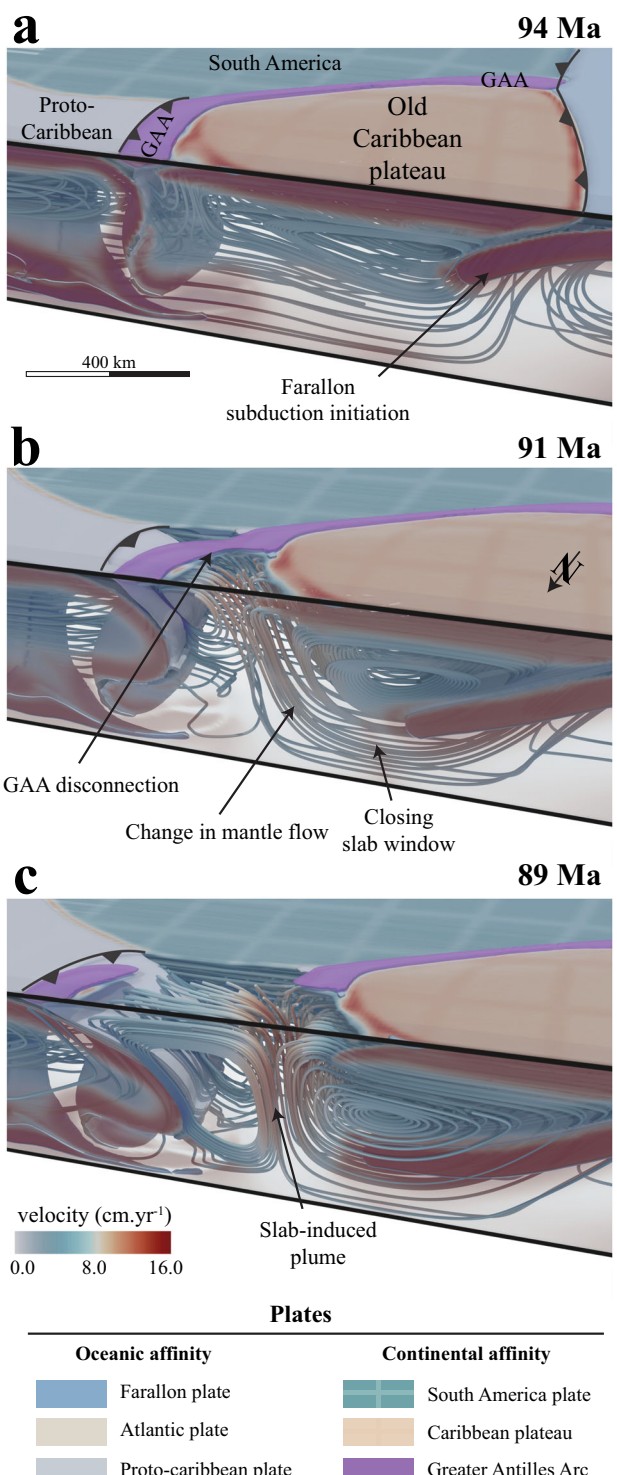

**Fig. 3 | Change in mantle flow during subduction initiation. a** Subduction initiation on the western edge of the Caribbean plateau. **b** Sinking of the Farallon slab below the plateau triggers upwelling of the mantle, back-arc opening and eastward drag of a segment of the Greater Antilles Arc, driven by subduction roll-back of the proto-Caribbean plate. **c** As the Farallon slab reaches the 660 km discontinuity, the plume is fully formed.

the time needed to initiate subduction on the western edge of the plateau and thus an earlier plume formation at ca. 98 Ma (see appendix 4.2 and simulation CP.7). This is an important result, which shows that second order features can have a significant control on the timing of the main tectono-magmatic events.

Although the heat input at the origin of the CLIP requires a lower mantle plume origin[10–12], it has also been proposed that a slab window and toroidal flow (Fig. 3b) might have supplied hot and dry astheno-sphere to mix with the wet mantle wedge, resulting in the CLIP for-mation under hydrous conditions[17]. However, others studies have shown that the geochemistry of the CLIP lavas is not compatible with a passive slab window environment[40] and that a slab window could not explain the cessation of the CLIP activity by 70 Ma[18]. The results of the simulation that account for adiabatic heating without accounting for a plume head (See simulation MP.1AH, supplementary Fig. 19) show that slab window and passive mantle flow alone is unable to generate either high enough mantle potential temperature or any significant hydrous partial melting. Given a plume-derived anomalously hot and buoyant mantle, we find that irrespective of the tested initial conditions, sub-duction initiation on the western edge of the plateau is able to trigger a key mantle flow reorganization against the rolling back proto-Caribbean slab. The resulting active closure of the slab window (Fig. 3b, c) drives the formation of a subduction-induced plume (Fig. 4), which is able to reach maximum temperature ≥ 1500 °C and generate up to 22 km thick crust (Fig. 2f). Moreover, we find that simulations with higher initial mantle temperature (see simulation MP.1c and MP.1d, Supplementary Figs. 15, 17) can predict maximal potential mantle temperature of ca. 1600 °C, close to the maximum potential mantle temperature recorded for the CLIP[10–12]. When scaled to realistic geodynamic timescale (see simulation MP.1c, supplemen-tary Fig. 15), we find that the calculated excess magma volume of $6.8 \times 10^6$ km³ is of the same order of magnitude compared to the estimated $4.4 \times 10^6$ km³ for the Caribbean plateau[34].

The accumulation of the stagnating proto-Caribbean slab (Fig. 2d) on the 660 km discontinuity may also have contributed to thermally destabilize the upper/lower mantle transition zone and thus further enhance the subduction-induced plume[41]. Such upper/lower mantle interactions are beyond the scope of this contribution and should be investigated in future studies which would include interaction with a plume generated at the core/mantle boundary, upper/lower mantle phase change and magma genesis/transfer. Whether or not the plume-head at the origin of the CLIP was solely captured during subduction-initiation, such as modeled here, or still actively ascending cannot be answered in this study. However, if a plume-head was still actively ascending at 95–70 Ma, the change of mantle flow related to renewal of Farallon subduction would probably have captured it through the slab window[42].

Despite the complexity of the Cretaceous Caribbean system, our simulations successfully account for a first-order suite of emerging geodynamic and tectonic processes such as mid-ocean ridge spread-ing, subduction initiation, back-arc spreading, trench propagation and mantle plume generation in a self-consistent manner. Our results thus propose for the first time a unifying Cretaceous geodynamic frame-work for the Caribbean region, including subduction-induced plume formation as a consequence of local plate and mantle flow reorgani-zation. While a similar scenario of subduction-driven magmatic pulse has been postulated as a possible explanation of intraplate off-volcanic arc-volcanism[43], our modelling results show that such a scenario is geodynamically feasible, and under plate-constrained conditions is able to induce plume formation, widespread mantle partial melting and the genesis of a large igneous province. Finally, our simulations further emphasize how the migration of subduction systems from a Pacific-type ocean into an Atlantic-type ocean can occur, showing that this is a feasible and likely mechanism to initiate new subduction zones in pristine oceans.

## Methods
### Mathematical formulation
The 3-D geodynamic simulations were performed using LaMEM (Kaus et al., 2016; https://bitbucket.org/bkaus/lamem). LaMEM is a finite

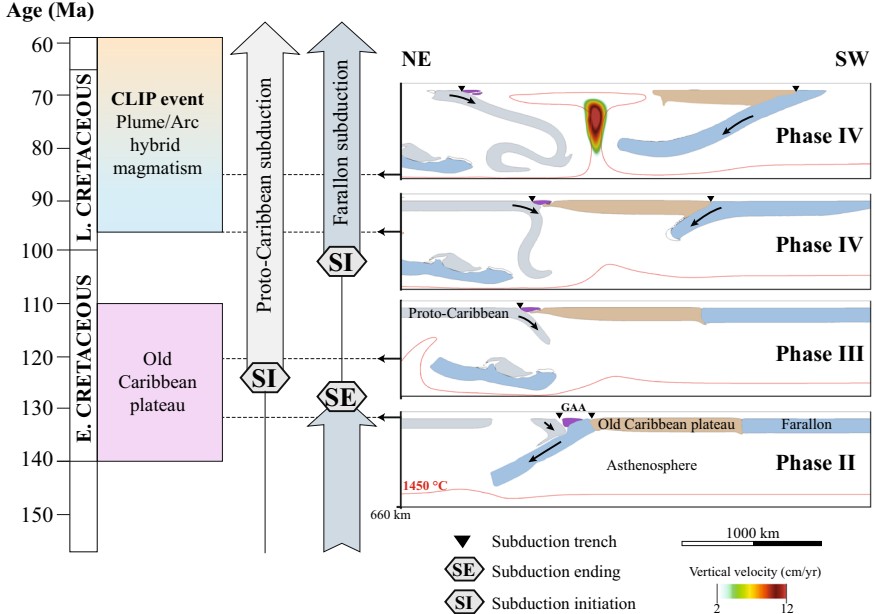

**Fig. 4 | Timeline of the main Cretaceous Caribbean tectono-magmatic events compared with the reference simulation.** The 4 cross-sections shown in this figure are slices performed 100 km away from the northern boundary of the 3D simulation. Subduction-initiation induced plume during phase IV is depicted by overlaying the vertical velocity field. Note that during phase IV the Caribbean plateau is thinning due to extensional forces as suggested by previous studies[11, 18].

Moreover, subduction renewal on the back of the plateau triggers a propagation of the plume-head below the Caribbean plate predominantly eastward, forcing shallow roll-back of the proto-Caribbean plate. Such forced roll-back is not observed in simulations for which the plume-head is ascending within a plateau aggregate (See simulation CP.7, Supplementary Fig. 12).

difference staggered grid discretization code that uses particle-in-cell methods to solve the energy, momentum and mass conservation equations. The rheologies of the rocks are assumed to be visco-elasto-plastic and the total deviatoric strain rate is given by:

$$\dot{\epsilon}_{ij} = \dot{\epsilon}_{ij}^{vis} + \dot{\epsilon}_{ij}^{el} + \dot{\epsilon}_{ij}^{pl} = \frac{1}{\eta_{eff}}\tau_{ij} + \frac{1}{2G}\frac{\partial \tau_{ij}}{\partial t} + \dot{\lambda}\frac{\partial Q}{\partial \sigma_{ij}}, \quad (1)$$

where $\dot{\epsilon}_{ij}^{vis} + \dot{\epsilon}_{ij}^{el} + \dot{\epsilon}_{ij}^{pl}$ are the viscous, elastic and plastic strain rates, respectively. $\eta_{eff}$ is the effective viscosity, $G$ the elastic shear modulus, $\tau_{ij}$ the deviatoric stress tensor, $t$ the time, $\dot{\lambda}$ is the plastic multiplier, $Q$ the plastic flow potential and $\sigma_{ij} = -P + \tau_{ij}$ the total stress. The effective viscosity $\eta_{eff}$ is given by:

$$\eta_{eff} = \frac{1}{2}A^{-\frac{1}{n}}\exp\left(\frac{E+PV}{nRT}\right)\dot{\epsilon}_{II}^{\frac{1}{n}-1}, \quad (2)$$

where $A$ is the exponential prefactor, $n$ the stress exponent of the dislocation creep, $\dot{\epsilon}_{II}$ is the second invariant of the viscous strain rate tensor, $E$, $V$ are the activation energy and volume, respectively, $P$ is the pressure, $R$ is the gas constant and $T$ is the temperature. Plasticity is modeled using the Drucker-Prager yield criterion given by:

$$\tau_y = \sin(\phi)P + \cos(\phi)C, \quad (3)$$

where $\tau_y$ is the yield stress, $\phi$ the friction angle and $C$ the cohesion. Strain softening is taken into account by linearly reducing both the friction angle and the cohesion of the material by a factor of 100 between 10 to 60% of accumulated strain. Minimum cohesion is set to 0.01 MPa and maximum yielding stress to 900 MPa.

The energy equation is solved as

$$\rho C_p \left(\frac{\partial T}{\partial t} + v_i \frac{\partial T}{\partial x_i}\right) = \frac{\partial}{\partial x_i}\left(k\frac{\partial T}{\partial x_i}\right) + H_S + H_A, \quad (4)$$

where $v_i$ is the velocity, $x_i$ the cartesian coordinates, $\rho$ the density, $C_p$ the heat capacity, $T$ the temperature, $t$ the time, $k$ the thermal conductivity, $H_S$ the shear heating and $H_A$ the adiabatic heating. The shear heating is defined as

$$H_S = \tau_{ij}\left(\dot{\epsilon}_{ij} - \dot{\epsilon}_{ij}^{el}\right), \quad (5)$$

and the adiabatic heating as

$$H_A = T\alpha g v_z \rho \quad (6)$$

where $g$ is the gravitational acceleration, $v_z$ the vertical velocity and $\alpha$ the thermal expansivity. All material densities are temperature and pressure dependent:

$$\rho = \rho_0 + \alpha(T - T_0) + \beta(P - P_0), \quad (7)$$

where $\rho_0$ is the density of reference of the material, $\beta$ is the compressibility and $P_0$ is the compressibility. For more detailed information about the LaMEM code, see Kaus et al. (2016).

## Model setup
The modeled region is restricted to the upper mantle with a size of $6000 \times 3000 \times 660$ km and a resolution of $384 \times 192 \times 96$ elements. Mechanical boundary conditions are set to be free-slip at the bottom, left, right, front and back walls, and, free surface at the top wall. Plume activity throughout most of Cretaceous times has been clearly demonstrated in the Caribbean region[11,13,14]. We, thus, set the mechanical bottom boundary condition to be free-slip in order to facilitate mantle upwelling from the upper/lower mantle interface. The initial plate configuration includes the South American continental plate, the oceanic proto-Caribbean, Atlantic and Farallon oceanic plates, the Greater Antilles Arc and the Caribbean plateau. The Greater Antilles Arc and South American crust are set to be 35 km thick, the oceanic crusts 15 km thick and the plateau crust 20 km thick.

**Table 1 | Model parameters**

| - | Mantle | Felsic crust | Oceanic crust | Weak zones |
|---|---|---|---|---|
| Flow law | Dry olivine[a] | Quartzite[b] | Dry olivine[a] | Diabase[c] |
| Pre-exp. factor (Pa$^{-n}$. s$^{-1}$) | $1.1 \times 10^5$ | $6.7 \times 10^{-6}$ | $1.1 \times 10^5$ | 8.0 |
| Activation energy (J.mol$^{-1}$) | $530 \times 10^3$ | $156 \times 10^3$ | $530 \times 10^3$ | $485 \times 10^3$ |
| Activation volume (m$^3$. mol$^{-1}$) | $11$–$16.0 \times 10^{-6}$ | 0.0 | $11$–$16.0 \times 10^{-6}$ | 0.0 |
| Stress exponent | 3.5 | 2.4 | 3.5 | 4.7 |
| Density (kg.m$^{-3}$) | 3240.0–3300.0 | 2800.0 | 3300.0 | 3300.0 |
| Thermal expansivity (K$^{-1}$) | $3 \times 10^{-5}$ | $3 \times 10^{-5}$ | $3 \times 10^{-5}$ | $3 \times 10^{-5}$ |
| Conductivity (W.mK$^{-1}$) | 3.0 | 3.0 | 3.0 | 3.0 |
| Heat capacity (J.K$^{-1}$) | 1050.0 | 1050.0 | 1050.0 | 1050.0 |
| Shear modulus (MPa) | $5 \times 10^{10}$ | $5 \times 10^{10}$ | $5 \times 10^{10}$ | $5 \times 10^{10}$ |
| Cohesion (MPa) | 30.0 | 30.0 | 5.0 | 30.0 |
| Cohesion softened (MPa) | 0.3 | 0.3 | 0.05 | 0.3 |
| Friction angle (°) | 10.0 | 10.0 | 0.0 | 1.0 |
| Friction angle softened (°) | 0.1 | 0.1 | 0.0 | 0.01 |

The minimum and maximum viscosities are capped to $10^{19}$ and $10^{23}$ Pa s, respectively.
[a]Hirth & Kohlstedt (2003).
[b]Ranalli (1995).
[c]Mackwell et al. (1998).

Surface temperature is fixed at 20 °C while bottom temperature is fixed at 1525 °C using an initial temperature gradient in the upper mantle of 0.3 °C.km$^{-1}$ (see Fig. 1c). Initial temperature profiles for the oceanic plates are prescribed according to the half-space cooling model[44] (Fig. 1b) using half-spreading rates of 1.5 cm.yr$^{-1}$ for the Atlantic and Farallon plates and 1.0 cm.yr$^{-1}$ for the proto-Caribbean plate. For the South American lithosphere the initial temperature profile is computed using a cooling age of 140 Myr and a lithosphere thickness of 160 km. The initial temperature profile for the old Caribbean plateau lithosphere uses a cooling age of 25 Myr and a lithosphere thickness of 110 km. Once the initial temperature profile has been computed for all oceanic and continental lithospheres, mantle material with temperature greater than 1250 °C is turned into asthenophere. This strategy allows to define the initial plate thickness as illustrated in Fig. 2. In order to allow for plume formation, we use the Boussinesq approximation in all simulations but MP.1AH and simulations MP.AH.HA.1 to 4 for which we use the extended Boussinesq approximation. The choice of using the Boussinesq approximation is supported by clear evidence of plume activity throughout the modeled time period (140–110 and 100–70 Ma)[11,13,14]. We employ non-linear visco-elasto-plastic rheologies with a set of parameters provided in Table 1. Subduction is pre-established along the western South American margin by a pre-subducted 300 km deep slab segment. In order to keep the air layer at 20 °C the conductivity in the air is artificially set to $= 100.0$ W.m.K$^{-1}$ and the heat capacity is set to $= 1.0 \times 10^6$ J.K$^{-1}$. The initial buoyancy constrast between this pre-subducted slab and the surrounding sub-continental mantle allow for self-sustained gravity-driven subduction. No internal constraints such as velocity or stress boundary conditions are imposed on any of the plates.

The 31 simulations presented here (Table 2) have been performed on the data center of the Johannes Gutenberg University Mainz: Mogon II. Each simulation has used 128 cores with a computational wall time of 28 h or was stopped when the simulation reached 1400 time-steps. The set of simulations can be divided into two main series. The first serie uses a single rectangular(ish) Caribbean plateau of variable dimensions (e.g., models MP.1 to MP.2), while the second investigate the role of a composite Caribbean plateaus made of aggregated sub-circular smaller plateaus separated by wide weak zones (e.g., models CP.1 to CP.4) or by oceanic crust (e.g., models CP.5 and CP.6). For both series the simulation time is scaled to fit as best as possible the real geodynamic time by varying the mantle activation volume in the range

of 11.0 to $16.5 \times 10^{-6}$ m$^3$. mol$^{-1}$. The control of the initial mantle temperature gradient is studied in simulations MP.1a to MP.1d. For these simulations, we explore an initial mantle temperature gradient of 0.2 °C.km$^{-1}$ (models MP.1a and MP.1b) and 0.4 °C.km$^{-1}$ (models MP.1c to MP.1f) with a fixed bottom temperature of 1460 and 1600 °C, respectively (see Fig. 1c). The simulation time for models MP.1a to MP.1f is also scaled by varying the mantle activation volume. This strategy allows to compare the estimated volume of excess magma in a relevant manner.

In order to compare the Boussinesq approximation against the extend Boussinesq approximation we performed 4 additional simulations using the extended Boussinesq approximation (adiabatic heating activated) and with a prescribed heat anomaly (anomalously hot mantle representing a plume head) initially placed underneath the old plateau (models AH.HA.1 to AH.HA.4, see Fig. 5 and Supplementary Figs. 20–23). In this anomalously hot region, the initial adiabatic gradient of 0.6 °C.km$^{-1}$ imposed elsewhere is elevated following a gradient of 0.8 °C.km$^{-1}$ between 400 and 660 km depth. This results in a maximum temperature at the bottom of the anomalously hot region reaching 1800 °C for models AH.HA.1 to AH.HA.4 (Fig. 5 and Supplementary Figs. 20–23).

The feasibility of a passive mantle upwell origin under hydrous conditions for the CLIP[17] is explored in simulation MP.1AH (see Table 2 and Supplementary Fig. 19) for which the extended Boussinesq approximation is also used (adiabatic heating on). Partial melt content was post-processed using the hydrous mantle melting parameterization[32] using a water content of 0.0 wt% in all simulations (dry peridotite) but simulation MP.1AH where a water content of 0.3 wt% is used instead, to investigate hydrous partial melting in passive upwelling conditions. The resulting igneous crust thickness is estimated by vertically integrating the computed melt fraction, while the total excess volume of magma is estimated by integrating the crust thickness greater than 10 km over the partially melting area. Note that the crust thickness and excess volume of magma are both estimated when the plume is fully formed.

It is worthwhile to emphasize that even though our 3D gravity-model simulations allow to capture the geodynamic evolution of a simplified Caribbean plate configuration, they do not take into account high order features such as phase change, magma genesis/transfer and small-scale inherited regional structures that can locally yield variations from the modeled geodynamic framework presented here. Although, higher order features are certainly relevant to account for

**Table 2 | Investigated parameters**

| Simulation | Plateau area (km²) | Mantle V | WZ IFA | Other |
|---|---|---|---|---|
| SP.1 | 427,925 | 13.50 | 2.0 | - |
| SP.2 | 433,735 | 14.00 | 2.0 | - |
| SP.3 | 433,735 | 14.50 | 2.0 | - |
| MP.1 | 776,598 | 14.25 | 1.0 | reference model |
| MP.1a | 776,598 | 11.00 | 1.0 | $\nabla T^{UM} = 0.2\,°C.km^{-1}$, bottom T = 1460 °C |
| MP.1b | 776,598 | 12.25 | 1.0 | $\nabla T^{UM} = 0.2\,°C.km^{-1}$, bottom T = 1460 °C |
| MP.1c | 776,598 | 15.25 | 1.0 | $\nabla T^{UM} = 0.4\,°C.km^{-1}$, bottom T = 1600 °C |
| MP.1d | 776,598 | 15.75 | 1.0 | $\nabla T^{UM} = 0.4\,°C.km^{-1}$, bottom T = 1600 °C |
| MP.1AH | 776,598 | 12.25 | 1.0 | Adiabatic heating, 0.3 wt% $H_2O$ |
| MP.AH.HA.1 | 776,598 | 16.75 | 1.0 | Adiabatic heating, Adiabatic gradient 0.6 °C, heat anomaly |
| MP.AH.HA.2 | 776,598 | 16.75 | 1.0 | Adiabatic heating, Adiabatic gradient 0.6 °C, wider heat anomaly |
| MP.AH.HA.3 | 776,598 | 16.25 | 1.0 | Adiabatic heating, Adiabatic gradient 0.6 °C, heat anomaly |
| MP.AH.HA.4 | 776,598 | 16.75 | 1.0 | Adiabatic heating, Adiabatic gradient 0.6 °C, heat anomaly |
| MP.2 | 776,598 | 14.25 | 1.0 | proto-Caribbean $\rho = 3290\ kg.m^{-3}$ |
| MP.3 | 776,598 & 2 blocks | 14.25 | 1.0 | - |
| MP.4 | 776,598 & 4 blocks | 14.25 | 1.0 | - |
| MP.5 | 776,598 | 14.50 | 1.0 | - |
| M2P.1 | 706,264 | 14.00 | 1.0 | - |
| M2P.2 | 706,264 | 14.25 | 1.0 | - |
| M3P.3 | 612,809 | 14.00 | 1.0 | - |
| LP.1 | 818,701 | 14.25 | 1.0 | - |
| LP.2 | 725,003 | 14.25 | 2.0 | - |
| CP.1 | 427,603 & 3 rounded WZ blocks | 14.25 | 1.0 | plateau V = 14.00 |
| CP.2 | 427,603 & 3 rounded WZ blocks | 14.25 | 1.0 | plateau V = 13.50 |
| CP.3 | 368,389 & 3 rounded WZ blocks | 14.50 | 1.0 | plateau V = 14.00 |
| CP.4 | 406,917 & 3 rounded WZ blocks | 14.25 | 1.0 | plateau V = 14.00 |
| CP.5 | 517,164 & 3 rounded blocks | 14.25 | 1.0 | - |
| CP.6 | 537,607 & 3 rounded blocks | 14.50 | 1.0 | - |
| CP.7 | 368,221 & 3 rounded WZ blocks | 14.25 | 1.0 | plateau V = 14.00, no N-S proto ridge |
| CP.8 | 399,372 & 3 rounded WZ blocks | 14.25 | 1.0 | plateau V = 14.00, no N-S proto ridge |

Results of all simulations are provided in the Supplementary Information.

*V* activation volume, *IFA* internal frictional angle, *WZ* weak zone, *UM* upper mantle.

(e.g., to explain regional variations and/or to further improve the accuracy of the modeled sequence of event with respect to plate reconstructions), they are not the main driver of the geodynamic evolution of the system. Instead, higher order features either form as a direct consequence of the geodynamic evolution (e.g., partial melting) or constitute a set of additional constraints (e.g., inherited structures). We are thus confident that our modelling approach allow comparing the model predictions with the first-order tectono-magmatic events of the Caribbean system (Fig. 4).

## Data availability
The input files, scripts and instructions to perform the simulations, process the simulation outputs, and produce the figures of this research have been deposited on Zenodo at: https://doi.org/10.5281/zenodo.7569683.

## Code availability
The version of the softwares used for this research have been deposited on Zenodo at LaMEM:[30], https://doi.org/10.5281/zenodo.7405012 and geomIO:[45], https://doi.org/10.5281/zenodo.7405022.

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

## Acknowledgements

This study was funded by the European Research Council through the MAGMA project, ERC Consolidator Grant #771143. Simulations of this research were conducted using the supercomputer Mogon II of the Johannes Gutenberg University Mainz (hpc.uni-mainz.de).

## Author contributions

N.R. designed the initial idea, performed the simulations and wrote the paper. J.C.D., J.A., F.R. and B.K. contributed to design the set of simulations. Y.R.-A. contributed to better constrain the modelling results with the geological record of the Caribbean. A.P. and B.K. wrote the geodynamic code used in this study. All authors participated in further discussions and revision of the manuscript.

## Funding

## Competing interests

The authors declare no competing interests.
