## [Peer Review File · Nature Communications]

Subduction initiation triggered the Caribbean Large Igneous ProvinceREVIEWER COMMENTS

Reviewer #1 (Remarks to the Author):

This is an interesting and timely paper with potentially strong impact, which proposes a unified geodynamic model for the Caribbean region evolution. However, the paper needs some improvements:

1. Logic is missing in some places (see specific comments) and some important statements made by the authors are incorrect (see specific comments).
2. Assumed initial thermal structure of the model has a key significance for the development of either passive (adiabatic) or active (hot plume) subduction-induced upwelling proposed in this paper. Influence of this thermal structure needs careful testing and discussion.
3. There is an inconsistency in developing of the model thermal structure. Mantle seems to be insufficiently cooled adiabatically. Temperature at the lower boundary is set at 1525 C and an adiabatic temperature gradient of 0.3 C/km is assumed. This is not consistent with 1450C isotherms rising to near model surface in Fig.2F. $T(60\text{km})=T(660\text{km})-0.3*600=1525-180=1305$ C, which is the hottest possible temperature at 60 km depth if the assumed adiabatic cooling is considered.

Specific comments for the paper are given below.

Taras Gerya, Zurich, 12.07.2022

Specific comments

Line 15. "The Earth's oceanic lithosphere is continuously being recycled at subduction zones but unless external forces come into play, new subduction zones are unlikely to spontaneously initiate. Instead, subduction zones are more likely to propagate from old oceans (Pacific-type) into pristine (Atlantic-type) oceans." Logic is missing here. Subduction zones propagation from old to new ocean is a disputable process, which is not an alternative to spontaneous subduction initiation. In fact, propagation is not initiation since no new subduction zone is created by this process. Many horizontally and vertically forced initiation mechanisms have been suggested on the basis of observation and models (e.g., reviews by Stern and Gerya, 2018; Cramer et al., 2020; Lalemand and Arcay, 2021).

Line 32. «Our modelling results thus better constrain the mechanism of subduction transfer in the Caribbean region and highlight a new geodynamic process able to generate Large Igneous Provinces without the need of a lower mantle rooted plume.» Can it also explain self-consistently high mantle potential temperature recorded in the Caribbean LIP (e.g., by Gorgona komatiites)?

Line 42. "While oceanic spreading and subduction dynamics are now fairly well understood, the transition from a passive spreading ocean to an active subduction zone together with its geodynamic consequences remain poorly investigated[1]. This is mainly because there is no evidence of ongoing spontaneous subduction initiation on Earth[2-4] and our rheological understanding of passive margins suggests that they are too strong to spontaneously break and initiate subduction[5]." The logic is weak here. Why spontaneous initiation (and not just initiation) should matter in this context? Ongoing subduction initiation has been suggested and studied for example for Puysegur trench (Gurnis et al., 2019). An intrinsic strength reduction of passive margins with time has also been suggested (Bercovici and Mulukova, 2020).

Line 51. "While voluminous magmatic and volcanic products are generally attributed to deep sourced mantle plumes, in the Caribbean major geodynamic inconsistencies remain." What inconsistencies? Short discussion and references needed.

Line 54. "Understanding the geodynamic conditions that led to the transfer of subduction from Pacific to Atlantic in the Caribbean region is therefore of primary importance, not only to unravel the formation of the Caribbean large igneous province AND the processes of subduction initiation, but also because it provides an enhanced comprehensive picture of modern plate tectonics." Not sure why "it provides an enhanced comprehensive picture of modern plate tectonics" and what is this picture...

Line 96. "Plume-related weakening of the lithosphere and edge downwelling has been proposed to

have triggered simultaneous subduction initiation beneath the old plateau of the adjacent proto-Caribbean and Farallon plates [8, 9, 18]. However, evidence shows that the proto-Caribbean subduction initiated 35 Ma earlier 101 at ca. 135 Ma [10, 13, 15].” This is not correct. Reference 8 (Whattam and Stern, 2014) suggests “Late cretaceous plume-induced subduction initiation along the southern margin of the Caribbean and NW South America”, which is not the proto-Caribbean subduction that existed before (see their Fig. 14).

Fig. 2 and all other model figures in this paper. Model dimensions/length scales are missing.

Fig. 2. Initial temperature structure of the model is only shown very coarsely but is a key model input. Showing vertical mantle temperature profile would be very useful.

Line 168 Meanwhile, subduction initiation of the Farallon subduction is accompanied by a major change of the proto-Caribbean underlying mantle flow (Fig. 3A,B). As the Farallon plate sinks below the Caribbean plateau, the closing mantle window forces the mantle to flow upward, effectively generating an upper-mantle scale plume (Fig. 3C) which we coin subduction-induced plume.” Plume is an active upwelling driven by own buoyancy. The thermal structure of the subduction induced upward flow will depend on the assumed initial thermal structure of the model. Adiabatic temperature profile should not create any thermal anomaly (and thus plume) but rather a forced passive upwelling without any temperature anomaly compared to the surrounding mantle. If initial mantle temperature profile assumed in this model deviates from the adiabatic (it seems to be a hot temperature boundary layer close to the lower boundary) – this needs an explanation since the lower boundary is at 660 km (not at CMB) in this model.

Line 199. “Using parameterized dry mantle melting [23], we find that partial melt content reaches a maximum of 40 wt% in the central region of the plume-head (Fig. 2F) consistent with geochemical estimates of 32 wt% [20, 26]. Calculation of the generated crust thickness ranges from 10 to 22 km and is also consistent with available data on the present-day Caribbean plateau crust thickness [16].”

Line 230. Here, we find that irrespective of the tested initial conditions, subduction initiation on the western edge of the plateau is able to trigger a key mantle flow reorganization against the rolling back proto-Caribbean slab. The resulting active closure of the slab window (Fig. 3B,C) drives the formation of a subduction-induced plume (Fig. 4) which is able to reach maximum temperature _ 1500 _ and generate up to 22 km thick crust (Fig. 2F).

This result should again be critically dependent on the initial thermal structure of the model, which needs discussion. Influence of the Initial thermal model structure needs some testing.

Line 243. “However, if the plume was still active at 95-70 Ma, the change of mantle flow related to renewal of Farallon subduction would probably have captured the plume-head through the slab window [31], thus contributing to the 97-70 Ma CLIP event.” In my opinion, this seems to be a way to get an active hot and buoyant subduction-induced upwelling, not passive adiabatic upwelling.

Line 285. “Mechanical boundary conditions are set to be free-slip at the bottom, left, right, front and back walls, and, free surface at the top wall.” This needs discussion for the lower boundary since it is at 660 km, not at CMB.

Line 296. “The initial temperature profile for the Caribbean plateau lithosphere uses a cooling age of 25 Ma and a lithosphere thickness of 110 km. Once the initial temperature profile has been computed for all oceanic and continental lithospheres, mantle material with temperature greater than 1250 C is turned into asthenosphere. This strategy allows to define the initial plate thickness as illustrated in figure 2. Thermal initial and boundary conditions should be discussed better including showing the initial mantle temperature profile.

Line 305. “The temperature is set at 20 C at the surface and follows an adiabatic gradient of 0.3 C/km reaching 1525 C at 660 km depth.” This is not consistent with 1450C isotherms rising to near model surface in Fig.2F. $T(60\text{km})=T(660\text{km})-0.3*600=1525-180=1305\text{ C}$, which is the hottest possible temperature at 60 km depth... Something is wrong here!

References

- Bercovici, D., Mulyukova, E., 2021. Evolution and demise of passive margins through grain mixing and damage. *PNAS*, v. 118, e2011247118;
- Cramer, F., Magni, V., Domeier, M., Shephard, G.E., Chotalia, K., Cooper, G., Eakin, C.M., Grima, A.G., Gürer, D., Király, A., Mulyukova, E., Peters, K., Robert, B., Thielmann, M. (2020) A transdisciplinary and community-driven database to unravel subduction zone initiation. *Nature Communications*, 11, 1-14.
- Lallemant, S., Arcay, D. (2021) Subduction initiation from the earliest stages to self-sustained subduction: Insights from the analysis of 70 Cenozoic sites. *Earth-Science Reviews*, 221, 103779.
- Gurnis, M., Van Avendonk, H., Gulick, S.P.S., Stock, J., Sutherland, R., Hightower, E., Shuck, B., Patel, J., Williams, E., Kardell, D., Herzig, E., Idini, B., Graham, K., Estep, J., Carrington, L., 2019. Incipient subduction at the contact with stretched continental crust: The Puysegur Trench. *Earth Planet. Sci. Lett.* 520, 212–219.
- Stern, R.J., Gerya, T. (2018) Subduction initiation in nature and models: A review. *Tectonophysics*, 746, 173-198.
- Whattam, S.A., Stern, R.J.: Late cretaceous plume-induced subduction initiation along the southern margin of the caribbean and nw south america: The first documented example with implications for the onset of plate tectonics. *Gondwana Research* 27(1), 38-63 (2015)

Reviewer #2 (Remarks to the Author):

Dear Editor,

I read the manuscript entitled "Subduction initiation triggered by the Caribbean Large Igneous Province", by Riel and co-authors with great interest.

The study is testing a previously published and valid geodynamical scenario (Pindell and other 2012) by the means of 3D high resolution numerical modeling. In the selected, scenario the entrance in the subduction zone of a buoyant feature (overthickened oceanic crust: Large Igneous Province-like) trigger subduction initiation at the Antilles trench. Models presented here demonstrate that the consequence of this subduction initiation is the rise of plume responsible for the emplacement of the CLIP (Caribbean Large Igneous Province). The models shown in this study are standing but there is a major pitfall in the manuscript: the link to Nature and the Geology of the Caribbean plate is not presented by the authors.

While the authors decipher the old Caribbean plate/Caribbean plateau formed while sitting in the downgoing Pacific Plate (sensu-lato) from the Caribbean Large Igneous Province, which, formed in situ as a result of Antilles (east dipping) subduction initiation. The manuscript crudely lacks of a map (which should stand in Fig. 1) synthesizing the present-day knowledge of the occurrences of these two geological units (age, methods on the age, error bar, nature and location of the samples). This map is of primary importance for the paper and must set the scene for the broad readership of Nature Communications. Therefore, based on the aforementioned argument, I would recommend you to reject the paper in its present form.

Reviewer #3 (Remarks to the Author):

Please see the attached review.

Scott A. Whattam

Department of Geosciences

King Fahd University of Petroleum and Minerals

Review of *Nature Communications* Manuscript NCOMMS-22-23100: *Subduction initiation triggered the Caribbean Large Igneous Province* by Riel et al.

By Scott A. Whattam 30-July-2022

Nature Communications Editor

Dear Handling Editor and Chief Editor,

I have now finished my review of the aforementioned *Nature Comm* Manuscript which I provide below.

MS DESCRIPTION & SYNOPSIS

On the basis of 3D geodynamic models, the authors propose a new tectonic model for the Caribbean Large Igneous Province (CLIP) that entails subduction induced plume formation.

PRESENTATION & SCIENTIFIC INTERPRETATION

The manuscript is very well written with few typos (ones spotted are given in Specific Points below) and was a pleasure to read. I am not a numerical modeler and so cannot comment on details of their methodology though the parameters used in the set-ups appear accurate (e.g., size, thickness of oceanic and plateau crust) appear correct. I do however, have many questions/comments regarding other aspects of the manuscript including inaccurate/misleading text and the use of inappropriate references which I outline in the Major Points below.

MAJOR POINTS

1. Abstract: Lines 15-17: this is not accurate according to many researchers (e.g., see Stern, 2003); moreover, this presents a paradox akin to the chicken and egg scenario as to which came first; how could the first subduction initiation (SI) event on Earth have formed if not spontaneously? In the absence of pre-existing plate tectonics (i.e., subduction), induced (i.e., collision caused by pre-existing lithospheric plates) SI could not have occurred
2. Lines 52-54: OK, perhaps geodynamic inconsistencies exist (but see Gerya et al., 2015; Stern and Gerya, 2018; Baes et al., 2021), but evidence of a mantle plume source in the case of the CLIP is undeniable based on a myriad of studies (e.g., Herzberg and Gazel, (2009) who showed potential Ts of 1560-1620 C and 1500 C at present; Whattam (2018) who showed same potential T of 1633 ±47 C for earliest plume-arc basalts; Gazel et al. (2021), who showed high 3He/4He signature in central Panama); perhaps this point is moot, as reading further it is apparent that the authors do not dispute a plume origin, but perhaps this should be revised to reflect the above
3. Lines 67-68: this is not entirely accurate as the CLIP is widely accessible (as shown in many studies) and is accessible in Panama, Costa Rica, the Lesser Antilles and western S. America; moreover, the CLIP has been drilled (e.g., Sinton et al., 1998, Leg 15 DSDP); the geochemistry, isotopic composition and (in part) the age is well constrained based on widespread accessibility
4. Lines 99-101: I know no evidence of “proto-Caribbean” subduction at 135 Ma; and please define what “proto-Caribbean” means
5. Lines 104-105: the authors appear to purport that the two plateaus (139-111 Ma and 100 Ma and younger) plateaus are distinct; this is inaccurate as Hoernle et al. (2004) show that the younger erupted through the older plateau as the Nicoya Complex shows ages of 139-11 Ma, whereas other ages here show younger ages of 90-84 Ma (Sinton et al., 1998); so, the authors proposed “collision of old plateau with the proto-Caribbean plate makes little sense
6. Line 117: add Herzberg and Gazel (2009) to the reference for “hot mantle source”
7. Line 118: what is meant by “geodynamically reconciled”?
8. Line 172: have any other studies proposed or shown “subduction-induced plume”?; this does not necessarily seem unreasonable, however, for this to have occurred, the slab would (likely) need to have first reached the mantle-core boundary (assuming plume origin here and partial melting of slab as source of the plume); what is the expected duration in m.y. expected/needed for the slab to reach the D”? Is this duration reasonable for the authors proposed models?
9. Line 188: Jolly et al. (2001) needs to be referenced here
10. Lines 190-192: tectonic reconstructions confirm subduction on western edge? What is the geochemical and geochronological data to substantiate this? The use of “confirm” is too strong-use suggest or perhaps infer instead; as well, Whattam and Stern (2015) show a hybrid subduction-plume signature at 100 Ma (e.g., as shown in NW Colombia, please

refer to appropriate refs. In Whattam and Stern, 2015), so how to explain the clear the undeniable plume isotope signature?

11. Line 211: Whattam and Stern (2015) do not provide any ages but cite studies that do and these latter ones are the ones that need be cited instead
12. Lines 223-224: note that there is debate as to whether Gorgona belongs to the CLIP (e.g., Kerr and Tarney, 2005)
13. Lines 242-243: well, the youngest age of the “old” CLIP is 111 Ma (Hoernle et al., 2004) so this suggests that the CLIP was not active between 111-100 Ma
14. In general: why do none of the reconstructions go back to 140 Ma?

SPECIFIC POINTS

1. Line 73: insert ‘the’ before Cocos
2. Line 75 and elsewhere: why are east and north capitalized?
3. Line 78: remove ‘an’
4. Line 159: typo (drag)
5. Lines 263-264: Acknowledgements need go before Supplementary information
6. Line 293: space needed between ‘mode’ and [33]
7. Pg. 14, Fig. B1: panels in these and subsequent figures should be arranged from top to bottom from old to young as in Figs. 2, 3

ANSWER TO REVIEWER NCOMMS-22-23100A

The reviewer's comments are in black while the answers to the reviewer's comments are in blue.

Reviewer #1 (Remarks to the Author):

This is an interesting and timely paper with potentially strong impact, which proposes a unified geodynamic model for the Caribbean region evolution. However, the paper needs some improvements:

1. Logic is missing in some places (see specific comments) and some important statements made by the authors are incorrect (see specific comments).

we agree and clarified the logic throughout the manuscript. Furthermore we clarified the statements that can be potentially wrongly interpreted following the reviewer's comments.

2. Assumed initial thermal structure of the model has a key significance for the development of either passive (adiabatic) or active (hot plume) subduction-induced upwelling proposed in this paper. Influence of this thermal structure needs careful testing and discussion.

We fully agree with the reviewer and did our best to improve and clarify this important point in the manuscript in several places (L. 145-151, 456-469). Here, we provide a short summary of the changes while they are fully detailed for each specific comment (discussed below).

In our simulations, we assume that the upper mantle of the Caribbean region is hotter than the ambient mantle (super-adiabatic) and thus able to generate subduction-induced plumes, consistent with the clear plume activity throughout Cretaceous times (Hoernle et al., 2004; Whattam & Stern, 2015; Whattam et al., 2018)(140-110 and 100-70 Ma). Numerically, we simulated this by employing the Boussinesq approximation and in addition having an increasing temperature with depth throughout the mantle. This aspect is now properly presented and explained throughout the manuscript (L...). To better explore the role of the initial thermal structure on the development of subduction-induced plumes, we performed 4 additional simulations using the Boussinesq approximation with colder and warmer initial temperature gradients in the upper mantle of 0.2 and 0.4 °C/km. Moreover, in order to validate that the simulations using the Boussinesq approximation yield similar results to simulations that employ the extended Boussinesq approximation, we present 4 additional simulations using the extended Boussinesq approximation. In these simulations, we prescribe an initial adiabatic temperature mantle gradient of 0.6 K/km and an initial super-adiabatic heat anomaly (plume head) placed underneath the old plateau. We show that the simulations using the extended Boussinesq approximation and accounting for an anomalously hot plume region, yields very similar results compared to the simulations using the Boussinesq approximation, which demonstrates that our conclusions are robust.

3. There is an inconsistency in developing of the model thermal structure. Mantle seems to be insufficiently cooled adiabatically. Temperature at the lower boundary is set at 1525 C and an adiabatic temperature gradient of 0.3 C/km is assumed. This is not consistent with 1450C isotherms rising to near model surface in Fig.2F. $T(60\text{km})=T(660\text{km})-0.3*600=1525-180=1305$ C, which is the hottest possible temperature at 60 km depth if the assumed adiabatic cooling is considered.

We agree that this part was unclear. In our modelling we indeed assumed that there was a superadiabatic temperature gradient of 0.3 K/km, which we now clarify (and double-check with additional simulations).

As the other reviewers also pointed out this is consistent with the clear evidence for active plume activity in the Caribbean region throughout Cretaceous times (Hoernle et al., 2004; Whattam & Stern, 2015; Whattam et al., 2018)(140-110 and 100-70 Ma).

Specific comments for the paper are given below.

Taras Gerya, Zurich, 12.07.2022

Specific comments

Line 15. “The Earth's oceanic lithosphere is continuously being recycled at subduction zones but unless external forces come into play, new subduction zones are unlikely to spontaneously initiate. Instead, subduction zones are more likely to propagate from old oceans (Pacif-type) into pristine (Atlantic-type) oceans.” Logic is missing here. Subduction zones propagation from old to new ocean is a disputable process, which is not an alternative to spontaneous subduction initiation. In fact, propagation is not initiation since no new subduction zone is created by this process. Many horizontally and vertically forced initiation mechanisms have been suggested on the basis of observation and models (e.g., reviews by Stern and Gerya, 2018; Cramer et al., 2020; Lallemand and Arcay, 2021).

we thank the reviewer for the helpful clarification. The sentence now reads: “While subduction is known to be the primary driving force of plate tectonics, the mechanisms leading to the formation of new subduction zones remain debated.” (L. 15)

Line 32. «Our modelling results thus better constrain the mechanism of subduction transfer in the Caribbean region and highlight a new geodynamic process able to generate Large Igneous Provinces without the need of a lower mantle rooted plume.» Can it also explain self-consistently high mantle potential temperature recorded in the Caribbean LIP (e.g., by Gorgona komatiites)?

The original simulations were indeed not able to predict such high mantle potential temperature. We performing additional simulations to investigate the role of initial thermal conditions, and find that the new simulations accounting for a higher super-adiabatic upper mantle temperature gradient of 0.4 °C/km are now able to predict maximum partial melting mantle temperature of ca. 1600°C, close to the maximum mantle temperature of 1630°C recorded for the CLIP event (Herzberg & Gazel, 2009; Whattam et al., 2018; Gazel et al., 2021). (L. 281-286)

Line 42. “While oceanic spreading and subduction dynamics are now fairly well understood, the transition from a passive spreading ocean to an active subduction zone together with its geodynamic consequences remain poorly investigated[1]. This is mainly because there is no evidence of ongoing spontaneous subduction initiation on Earth[2-4] and our rheological understanding of passive margins suggests that they are too strong to spontaneously break and initiate subduction[5].” The logic is weak here. Why spontaneous initiation (and not just initiation) should matter in this context?

we removed the reference to spontaneous subduction. (L. 42)

Ongoing subduction initiation has been suggested and studied for example for Puysegur trench (Gurnis et al., 2019). An intrinsic strength reduction of passive margins with time has also been suggested (Bercovici and Mulyukova, 2020).

we agree and corrected the sentence accordingly. The sentence now reads: “This is mainly because there is very limited examples of ongoing subduction initiation on Earth (Gurnis et al., 2019) and our

rheological understanding of passive margins suggests that they remain difficult to break and initiate subduction (Cloetingh et al., 1989; Mueller et al., 1991; Gurnis et al., 2004; Crameri et al., 2020) unless mechanical weakening of the passive margin is achieved (Bercovici & Mulyukova, 2021) and/or external forces are applied (Gurnis et al., 2019; Almeida et al., 2022)” (L. 42-46)

Line 51. “While voluminous magmatic and volcanic products are generally attributed to deep sourced mantle plumes, in the Caribbean major geodynamic inconsistencies remain.” What inconsistencies? Short discussion and references needed.

we agree and clarified this part of the text by pointing out that although it is now clearly established that “the late Cretaceous voluminous magmatic and volcanic activity recorded in the Caribbean (CLIP) is related to mantle plume activity with potential mantle temperature reaching up to 1630°C (Herzberg & Gazel, 2009; Whattam et al., 2018; Gazel et al., 2021), the relationships between the plume formation and the subduction dynamics of the Caribbean region remain poorly understood.” (L. 50-56)

Line 54. “Understanding the geodynamic conditions that led to the transfer of subduction from Pacific to Atlantic in the Caribbean region is therefore of primary importance, not only to unravel the formation of the Caribbean large igneous province AND the processes of subduction initiation, but also because it provides an enhanced comprehensive picture of modern plate tectonics.” Not sure why “it provides an enhanced comprehensive picture of modern plate tectonics” and what is this picture...

we agree and removed this part of sentence. The sentence now reads: “Understanding the geodynamic conditions that led to the transfer of subduction from Pacific to Atlantic in the Caribbean region is therefore of primary importance to unravel the formation of the Caribbean large igneous province and related subduction initiation” (L. 57-61)

Line 96. “Plume-related weakening of the lithosphere and edge downwelling has been proposed to have triggered simultaneous subduction initiation beneath the old plateau of the adjacent proto-Caribbean and Farallon plates[8, 9, 18]. However, evidence shows that the proto-Caribbean subduction initiated 35 Ma earlier 101 at ca. 135 Ma[10, 13, 15].” This is not correct. Reference 8 (Whattam and Stern, 2014) suggests “Late cretaceous plume-induced subduction initiation along the southern margin of the caribbean and NW South America”, which is not the proto-Caribbean subduction that existed before (see their Fig. 14).

we thank the reviewer for spotting this mistake. This part has been corrected and now reads as: “Subsequently, the plume head at the origin of the CLIP event was emplaced through the old plateau at ca. 100 Ma. Plume-related weakening of the lithosphere and edge downwelling has been proposed to have initiated north-east-directed subduction of the Farallon plate below the western to south-eastern edge of the Caribbean plateau” (L. 110-117)

Fig. 2 and all other model figures in this paper. Model dimensions/length scales are missing.

this has been added on all figures including supplementary figures.

Fig. 2. Initial temperature structure of the model is only shown very coarsely but is a key model input. Showing vertical mantle temperature profile would be very useful.

we agree with the reviewer and added panel C on figure 1 to show the initial temperature gradient.

Line 168 Meanwhile, subduction initiation of the Farallon subduction is accompanied by a major change of the proto-Caribbean underlying mantle flow (Fig. 3A,B). As the Farallon plate sinks below the Caribbean plateau, the closing mantle window forces the mantle to flow upward, effectively generating an upper-mantle scale plume (Fig. 3C) which we coin subduction-induced plume.” Plume is an active upwelling driven by own buoyancy. The thermal structure of the subduction induced upward flow will depend on the assumed initial thermal structure of the model. Adiabatic temperature profile should not create any thermal anomaly (and thus plume) but rather a forced passive upwelling without any temperature anomaly compared to the surrounding mantle. If initial mantle temperature profile assumed in this model deviates from the adiabatic (it seems to be a hot temperature boundary layer close to the lower boundary) – this needs an explanation since the lower boundary is at 660 km (not at CMB) in this model.

we thank the reviewer to help us clarifying this key part of the modeling strategy that was not detailed before. A paragraph has been added (L 145-151) “Based on clear evidence of long-lasting plume activity recorded in the Caribbean region throughout most of the Cretaceous (Hoernle et al., 2004; Whattam & Stern, 2015; Whattam et al., 2018)(140-110 and 100-70 Ma) that will likely have resulted in a super-adiabatic temperature structure, we employed the Boussinesq approximation along with an increased temperature gradient in our simulations i.e., we assume that the mantle of the Caribbean region was anomalously hot and buoyant due to plume activity during the entire modeled time window (140-70 Ma).”

Line 199. “Using parameterized dry mantle melting[23], we find that partial melt content reaches a maximum of 40 wt% in the central region of the plume-head (Fig. 2F) consistent with geochemical estimates of 32 wt%[20, 26]. Calculation of the generated crust thickness ranges from 10 to 22 km and is also consistent with available data on the present-day Caribbean plateau crust thickness[16].”

Line 230. Here, we find that irrespective of the tested initial conditions, subduction initiation on the western edge of the plateau is able to trigger a key mantle flow reorganization against the rolling back proto-Caribbean slab. The resulting active closure of the slab window (Fig. 3B,C) drives the formation of a subduction-induced plume (Fig. 4) which is able to reach maximum temperature ~ 1500 and generate up to 22 km thick crust (Fig. 2F).

This result should again be critically dependent on the initial thermal structure of the model, which needs discussion. Influence of the Initial thermal model structure needs some testing.

we fully agree with the reviewer and performed more than 24 simulations changing the initial, super-adiabatic, temperature gradient of the asthenosphere, and the used Boussinesq approximation (see details below). Among the 24+ new simulations we selected 8 new models presented in the supplementary material, for which the timing was more consistent. We now investigate the role of having an initial temperature gradient in the asthenosphere of 0.2 and 0.4 °C/km (on top of the 0.3°C/km used in the original models). As changing the temperature profile of the models has a major control on the rheology of the system, we varied the activation volume of the mantle accordingly in order to scale the modeling time to relevant geodynamic times comparable to the Caribbean system. A new paragraph “Role of the initial mantle temperature” has been added to the supplementary data.

Results show that a lower initial super-adiabatic temperature gradient of the upper mantle (0.2 °C/km) yields a significantly lower degree of partial melting with no predicted excess magma volume for simulation MP.1b (simulation with geodynamic time comparable to the Caribbean system). A higher gradient for the upper mantle (0.4 °C/km) yields higher mantle temperature of 1600 °C similar to the maximum mantle temperature recorded for the CLIP (ca. 1630°C, Herzberg & Gazel, 2009; Whattam et al., 2018; Gazel et al., 2021). Moreover, we find that for simulation MP.1c (simulation with

geodynamic time comparable to the Caribbean system) the predicted excess magma volume of $6.7 \times 10^6 \text{ km}^3$ is on the same order of magnitude to the estimated value of $4.4 \times 10^6 \text{ km}^3$ (Kerr et al., 2004). We also added a figure comparing the excess magma volume calculation for all simulations (Fig B27).

Line 243. “However, if the plume was still active at 95-70 Ma, the change of mantle flow related to renewal of Farallon subduction would probably have captured the plume-head through the slab window[31], thus contributing to the 97-70 Ma CLIP event.” In my opinion, this seems to be a way to get an active hot and buoyant subduction-induced upwelling, not passive adiabatic upwelling.

we agree with the reviewer and reflected this point across the manuscript. See previous comment on modeling assumption. Moreover, we performed an additional simulation for which adiabatic heating is activated, but with a purely adiabatic initial thermal mantle gradient. This model results further support the studies that shows that passive upwelling of the mantle (without super-adiabatic conditions) is unable to generate sufficient partial melting (under wet conditions) or even high enough potential mantle temperature. (Fig. B20)

Line 285. “Mechanical boundary conditions are set to be free-slip at the bottom, left, right, front and back walls, and, free surface at the top wall.” This needs discussion for the lower boundary since it is at 660 km, not at CMB.

We agree with the reviewer and added a discussion on the choice of free slip for bottom wall of the models. “Plume activity throughout most of Cretaceous times has been clearly demonstrated in the Caribbean region (Hoernle et al., 2004; Whattam & Stern, 2015; Whattam et al., 2018). We thus, set the mechanical bottom boundary condition to be free-slip in order to facilitate mantle upwelling from the bottom wall of the model i.e., at the upper/lower mantle transition.” (L. 343-346)

Line 296. “The initial temperature profile for the Caribbean plateau lithosphere uses a cooling age of 25 Ma and a lithosphere thickness of 110 km. Once the initial temperature profile has been computed for all oceanic and continental lithospheres, mantle material with temperature greater than 1250 C is turned into asthenosphere. This strategy allows to define the initial plate thickness as illustrated in figure 2. Thermal initial and boundary conditions should be discussed better including showing the initial mantle temperature profile.

we thank the reviewer for the helpful comment. We improved the description of thermal initial and boundary conditions. (L. 352-354) “Surface temperature is fixed at 20 °C while bottom temperature is fixed at 1525 °C using an initial super-adiabatic temperature gradient in the upper mantle of 0.3 °C/km (see Fig. 1C).” and (L. 390-396) “The control of the initial mantle temperature gradient is studied in simulations MP.1a to MP.1d. For these simulations, we explore an initial super-adiabatic mantle temperature gradient of 0.2 °C (models MP.1a and MP.1b) and 0.4 °C/km (models MP.1c to MP.1d) with a fixed bottom temperature of 1460 and 1600 °C, respectively (see Fig. 1C).”

Line 305. “The temperature is set at 20 C at the surface and follows an adiabatic gradient of 0.3 C/km reaching 1525 C at 660 km depth.” This is not consistent with 1450C isotherms rising to near model surface in Fig.2F. $T(60\text{km})=T(660\text{km})-0.3 \times 600=1525-180=1305 \text{ C}$, which is the hottest possible temperature at 60 km depth... Something is wrong here!

we agree with the reviewer and now clarify this. (L. 365-367): “In order to allow for plume formation, we impose a super-adiabatic gradient in all simulations except simulation MP.1AH. This choice is

supported by clear evidence of plume activity throughout the modeled time period (140-110 and 100-70 Ma) (Hoernle et al., 2004; Whattam & Stern, 2015; Whattam et al., 2018). In order to test the robustness of our conclusions, we performed additional simulations using the extended Boussinesq approximation which show that similar results are obtained if adiabatic effects are accounted for.”

References

- Bercovici, D., Mulyukova, E., 2021. Evolution and demise of passive margins through grain mixing and damage. *PNAS*, v. 118, e2011247118;
- Cramer, F., Magni, V., Domeier, M., Shephard, G.E., Chotalia, K., Cooper, G., Eakin, C.M., Grima, A.G., Gürer, D., Király, A., Mulyukova, E., Peters, K., Robert, B., Thielmann, M. (2020) A transdisciplinary and community-driven database to unravel subduction zone initiation. *Nature Communications*, 11, 1-14.
- Lallemand, S., Arcay, D. (2021) Subduction initiation from the earliest stages to self-sustained subduction: Insights from the analysis of 70 Cenozoic sites. *Earth-Science Reviews*, 221, 103779.
- Gurnis, M., Van Avendonk, H., Gulick, S.P.S., Stock, J., Sutherland, R., Hightower, E., Shuck, B., Patel, J., Williams, E., Kardell, D., Herzig, E., Idini, B., Graham, K., Estep, J., Carrington, L., 2019. Incipient subduction at the contact with stretched continental crust: The Puysegur Trench. *Earth Planet. Sci. Lett.* 520, 212–219.
- Stern, R.J., Gerya, T. (2018) Subduction initiation in nature and models: A review. *Tectonophysics*, 746, 173-198.
- Whattam, S.A., Stern, R.J.: Late cretaceous plume-induced subduction initiation along the southern margin of the Caribbean and NW South America: The first documented example with implications for the onset of plate tectonics. *Gondwana Research* 27(1), 38-63 (2015)

Reviewer #2 (Remarks to the Author):

Dear Editor,

I read the manuscript entitled “Subduction initiation triggered by the Caribbean Large Igneous Province”, by Riel and co-authors with great interest.

The study is testing a previously published and valid geodynamical scenario (Pindell and others 2012) by the means of 3D high resolution numerical modeling. In the selected scenario the entrance in the subduction zone of a buoyant feature (overthickened oceanic crust: Large Igneous Province-like) triggers subduction initiation at the Antilles trench. Models presented here demonstrate that the consequence of this subduction initiation is the rise of a plume responsible for the emplacement of the CLIP (Caribbean Large Igneous Province). The models shown in this study are standing but there is a major pitfall in the manuscript: the link to Nature and the Geology of the Caribbean plate is not presented by the authors.

While the authors decipher the old Caribbean plate/Caribbean plateau formed while sitting in the downgoing Pacific Plate (*sensu lato*) from the Caribbean Large Igneous Province, which formed in situ as a result of Antilles (east dipping) subduction initiation. The manuscript crudely lacks a map (which should stand in Fig. 1) synthesizing the present-day knowledge of the occurrences of these two geological units (age, methods on the age, error bar, nature and location of the samples). This map is of primary importance for the paper and must set the scene for the broad readership of *Nature Communications*. Therefore, based on the aforementioned argument, I would recommend you to reject the paper in its present form.

we thank the reviewer for this helpful suggestion and compiled the present-day knowledge of the different units: plateau, oceanic and arc-related in figure 1A. The addition is modified after Whattam & Stern, 2014. The range of ages of the different units are now displayed on the side panel of the figure and the complete list of references is provided in the figure caption. We believe that this may make our work even more appealing to the broad community.

Reviewer #3 (Remarks to the Author):

PRESENTATION & SCIENTIFIC INTERPRETATION

The manuscript is very well written with few typos (ones spotted are given in Specific Points below) and was a pleasure to read. I am not a numerical modeler and so cannot comment on details of their methodology though the parameters used in the set-ups appear accurate (e.g., size, thickness of oceanic and plateau crust) appear correct. I do however, have many questions/comments regarding other aspects of the manuscript including inaccurate/misleading text and the use of inappropriate references which I outline in the Major Points below.

MAJOR POINTS

1. Abstract: Lines 15-17: this is not accurate according to many researchers (e.g., see Stern, 2003); moreover, this presents a paradox akin to the chicken and egg scenario as to which came first; how could the first subduction initiation (SI) event on Earth have formed if not spontaneously? In the absence of pre-existing plate tectonics (i.e., subduction), induced (i.e., collision caused by pre-existing lithospheric plates) SI could not have occurred

we fully agree with the reviewer, and modified the beginning of the abstract to avoid this pitfall. The sentence now reads: “While subduction is known to be the primary driving force of plate tectonics, the mechanisms leading to the formation of new subduction zones remain highly debated.” (L. 15-18)

2. Lines 52-54: OK, perhaps geodynamic inconsistencies exist (but see Gerya et al., 2015; Stern and Gerya, 2018; Baes et al., 2021), but evidence of a mantle plume source in the case of the CLIP is undeniable based on a myriad of studies (e.g., Herzberg and Gazel, (2009) who showed potential Ts of 1560-1620 C and 1500 C at present; Whattam (2018) who showed same potential T of 1633 ±47 C for earliest plume-arc basalts; Gazel et al. (2021), who showed high ³He/⁴He signature in central Panama); perhaps this point is moot, as reading further it is apparent that the authors do not dispute a plume origin, but perhaps this should be revised to reflect the above

we fully agree with the reviewer that only a deep sourced plume can explain such high potential mantle temperature, and our models were consistent with this point of view. We have made this more explicit and corrected the manuscript to clarify this point (e.g., L. 52-56, 144-151).

3. Lines 67-68: this is not entirely accurate as the CLIP is widely accessible (as shown in many studies) and is accessible in Panama, Costa Rica, the Lesser Antilles and western S. America; moreover, the CLIP has been drilled (e.g., Sinton et al., 1998, Leg 15 DSDP); the geochemistry, isotopic composition and (in part) the age is well constrained based on widespread accessibility

we removed the sentence referring to the lack of access of the plateau.

4. Lines 99-101: I know no evidence of “proto-Caribbean” subduction at 135 Ma;

we agree with the reviewer. We wanted to say that subduction initiation is suggested to happen at 135 Ma and not that it was fully established. We clarified the sentence by stating that “Several studies however suggest that the westward subduction of the proto-Caribbean plate initiated 35 Ma earlier, at ca. 135 Ma (Jolly et al., 2001, Hastie et al., 2010a, Boschman 2014, Rojas et al., 2016, Cardenas et al., 2017, Torro et al., 2017, Boschman et al., 2019 and Rui et al., 2022)”. (L. 105-108)

and please define what “proto-Caribbean” means

we added the definition of the “proto-Caribbean” plate “as the former oceanic plate (Atlantic-derived) separating North and South America continental plates” (L. 93-95)

5. Lines 104-105: the authors appear to purport that the two plateaus (139-111 Ma and 100 Ma and younger) plateaus are distinct; this is inaccurate as Hoernle et al. (2004) show that the younger erupted through the older plateau as the Nicoya Complex shows ages of 139-11 Ma, whereas other ages here show younger ages of 90-84 Ma (Sinton et al., 1998); so, the authors proposed “collision of old plateau with the proto-Caribbean plate makes little sense

we agree that the two plateaus ultimately form only one plateau, with the CLIP emplaced through the old one. We were referring to the distinct event in time. We clarified this point by adding that the CLIP was emplaced through the old plateau (L. 97)

6. Line 117: add Herzberg and Gazel (2009) to the reference for “hot mantle source”

we added the reference

7. Line 118: what is meant by “geodynamically reconciled”?

we agree that this part of the sentence perhaps lacked logic and removed it. (L. 134)

8. Line 172: have any other studies proposed or shown “subduction-induced plume”?; this does not necessarily seem unreasonable, however, for this to have occurred, the slab would (likely) need to have first reached the mantle-core boundary (assuming plume origin here and partial melting of slab as source of the plume); what is the expected duration in m.y. expected/needed for the slab to reach the D”? Is this duration reasonable for the authors proposed models?

we fully agree with the reviewer. This point has been clarified throughout the manuscript. We now clearly state (L. 145-151, 456-469) that our modeling strategy assumes that the Caribbean mantle throughout the Cretaceous is anomalously hot and buoyant due to plume activity (i.e., in super-adiabatic conditions). The polarity change triggered a new plume-pulse in the upper mantle.

9. Line 188: Jolly et al. (2001) needs to be referenced here

reference has been added

10. Lines 190-192: tectonic reconstructions confirm subduction on western edge? What is the

geochemical and geochronological data to substantiate this? The use of “confirm” is too strong-use suggest or perhaps infer instead;

we agree and now use “infer” instead of “confirm”

as well, Whattam and Stern (2015) show a hybrid subduction-plume signature at 100 Ma (e.g., as shown in NW Colombia, please refer to appropriate refs. In Whattam and Stern, 2015), so how to explain the clear the undeniable plume isotope signature?

we thank the review for the helpful comment. We fully agree that the source of the CLIP is sourced from a deep plume. We updated the manuscript accordingly to reflect that in several places (L. 145-151, 456-469).

11. Line 211: Whattam and Stern (2015) do not provide any ages but cite studies that do and these latter ones are the ones that need be cited instead

we removed the citation, and added citations of the proper authors: Sinton et al., 1997; 1998; Hauff et al., 2000a; Revillon et al., 2000b

12. Lines 223-224: note that there is debate as to whether Gorgona belongs to the CLIP (e.g., Kerr and Tarney, 2005)

we were not aware of this contribution, and removed the reference to the Gorgona. We now instead refer to the references suggested by the reviewer on the high potential mantle temperature > 1500 °C (e.g., Herzberg and Gazel, 2009; Whattam et al., 2018; Gazel et al., 2021)(L. 260-265)

13. Lines 242-243: well, the youngest age of the “old” CLIP is 111 Ma (Hoernle et al., 2004) so this suggests that the CLIP was not active between 111-100 Ma

we agree with the reviewer and now discuss about plume-head activity and do not relate it with the old plateau plume activity.

14. In general: why do none of the reconstructions go back to 140 Ma?

we suppose that the plate reconstructions studies mainly focused on better understanding the CLIP plate tectonic framework while not focusing as much on the early Cretaceous period.

SPECIFIC POINTS

1. Line 73: insert ‘the’ before Cocos
done

2. Line 75 and elsewhere: why are east and north capitalized?
corrected

3. Line 78: remove ‘an’
done

4. Line 159: typo (drag)

corrected

5. Lines 263-264: Acknowledgements need go before Supplementary information
modified accordingly

6. Line 293: space needed between 'mode' and [33]
as this is a reference, we are not sure it needs a space

7. Pg. 14, Fig. B1: panels in these and subsequent figures should be arranged from top to bottom from old to young as in Figs. 2, 3

all the figures in the supplementary material have been re-arranged accordingly.

REVIEWERS' COMMENTS

Reviewer #1 (Remarks to the Author):

The paper has been improved by revisions, which also included running of important additional experiments. The only problem I found is a typo in Eq.6, where "/Cp" term is not needed. Otherwise the paper is in a good shape and can be recommended for publication.

Taras Gerya, Zurich, 04.01.2023

Final Revisions (NCOMMS-22-23100A)

Reviewer #1 (Remarks to the Author):

The paper has been improved by revisions, which also included running of important additional experiments. The only problem I found is a typo in Eq.6, where $"/C_p"$ term is not needed. Otherwise the paper is in a good shape and can be recommended for publication.

Taras Gerya, Zurich, 04.01.2023

Equation has been corrected.